# Fraction Execution Resolver Using a Hybrid Multi-CPU/GPU Encoding Scheme

**Georgios I. Papaioannou** [1] , **Maria Koziri** [2,*] , **Thanasis Loukopoulos** [1] **and Ioannis Anagnostopoulos** [1]

[1] Department of Computer Science and Biomedical Informatics, University of Thessaly, 35131 Lamia, Greece; geopapaioannou@uth.gr (G.I.P.); luke@dib.uth.gr (T.L.); janag@dib.uth.gr (I.A.)

[2] Department of Informatics and Telecommunications, University of Thessaly, 35131 Lamia, Greece

\* Correspondence: mkoziri@uth.gr

**Abstract:** Modern video coding standards make use of sub-pixel motion estimation to improve the video quality and reduce the bitrate. It is known that the fraction motion estimation (FME) part follows the integer motion estimation (IME) and adds an extra computational overhead due to the interpolation and the additional motion searches. In this paper, we propose a fraction execution resolver (FER) algorithm that lets the encoder skip the fraction part when specific criteria are met by introducing a preliminary fast test decision point (pFTDP) function for the IME part. If the pFTDP returns zero motion vectors (MVs) and the displacement search area center is also zero, then the fraction part is skipped. The pFTDP decision maker is executed only once, when a 2N × 2N block is first met, while all subsequent blocks follow this initial decision either by receiving the necessary MVs and RD from the pFTDP function or by using the precalculated IME values from the GPU kernel. For our experiments, we use a multithreaded CPU environment that also makes use of GPUs only for the integer part. Our evaluations provide a greater than 1600% encoding time saving at its peak in comparison with the default HEVC sequential mode and ideally a saving of greater than 2286% for still video frame sequences. The total average speedup for both Class A and Class B video sequences is ×13.45. The gain of the FER itself is more than ×3.9 compared with the same multithreaded setup environment. The PSNR and bitrate overhead observed are proportional to the tiling scheme used and are more related to the way CABAC works internally. The FER's negative effects on coding efficiency are proven to be negligible. A balance between speed and quality achieved by using a lower tiling pattern is shown to minimize the negative effects of the encoding scheme pattern. The experimental results confirm the validity of our motivation, namely, that we can benefit from a software fraction execution resolver without any extra hardware costs. The gain is further increased when video sequences have more static blocks than others.

**Keywords:** tile parallelism; fraction motion estimation; integer motion estimation; motion estimation; encoding complexity

## 1. Introduction

Video coding is a technique for compressing video data to minimize storage capacity and network bandwidth. As is known, the most intensive parts of any modern video encoder are the motion estimation (ME) calculations. Motion estimation includes integer-pixel ME and fraction-pixel ME. Typically, integer motion estimation (IME) is initially performed, and the best motion vectors from this stage are passed to the fraction motion estimation (FME) to achieve greater accuracy [1]. Fraction motion estimation is a two-step process. It first achieves half-pixel precision over integer pixel estimation by testing eight neighboring search points and then achieves quarter-pixel precision (Figure 1). A total of 16 search points are tested during the FME process, and the point with the minimum rate distortion (RD) is selected as the best candidate. The fraction calculations (Equation (1)) are undeniably another

CPU-intensive job for the motion estimation part, consuming more than 16% of the total time motion estimation (ME) needs according to our benchmark profiler.

$$
\begin{cases}
f\left(\overrightarrow{MV}_{pre}, \lambda_m\right) = SATD\left(\overrightarrow{MV}_{cur}, \quad \overrightarrow{MV}_{pre}\right) + \lambda_m \cdot R\left(\overrightarrow{MV}_{pre} - \overrightarrow{MV}_{cur}\right) \\
\overrightarrow{MV}_{best} = \underset{\overrightarrow{MV}_n}{\operatorname{argmin}} \, f\left(\overrightarrow{MV}_n, \lambda_m\right), \quad \overrightarrow{MV}_n \in \delta
\end{cases}
\tag{1}
$$

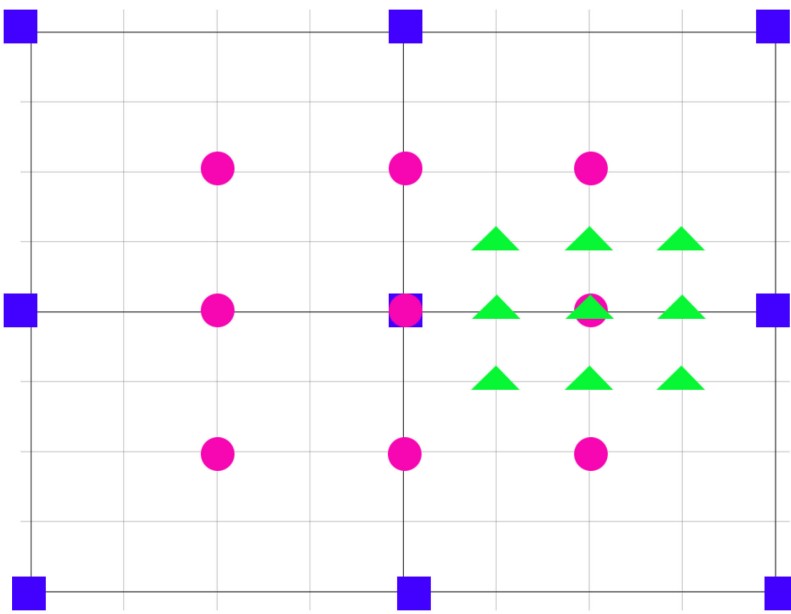

**Figure 1.** The HEVC FME process. Square symbols represent the integer pixel search points; the circles denote the interpolated half-pixel search points (1st step of the FME) and triangles the interpolated quarter-pixel search points (2nd step of the FME).

The SATD (sum of the transformed differences) measures the distortion when encoding the current PU ($\overrightarrow{MV}_{cur}$) with predictive motion vector c($\overrightarrow{MV}_{pre}$); $\lambda_m$ is the Lagrange multiplier; $R(\overrightarrow{MV}_{pre} - \overrightarrow{MV}_{cur})$ is the code bits required for the encoding; and $\delta$ is the motion vector candidates set for the FPME search points. FME searches among 16 neighboring candidate points ($\delta$) around the best IME motion vectors. The best candidate is the point with the minimum RD cost.

Fraction motion estimation typically follows integer motion estimation even though, in some cases, the extra computational efforts do not contribute extra accuracy to the already calculated motion vectors from the IME part. Hence, several software and hardware solutions have been published to minimize the FME computational load and thus minimize the total encoding times. Many of them include specialized hardware such as field-programmable gate arrays (FPGAs), while others are focused only on optimized algorithms at the software level. FPGAs are expensive and oriented toward companies and professionals that mostly need real-time video-encoding hardware [2–5]. In contrast, optimized software algorithms have been introduced to reduce the computational load [6–8] in variants of existing software video encoders without any additional costs. The proposed work in this paper is also a software algorithm that extends our proven hybrid encoder [9,10] to increase the speedup times by minimizing the calculations of the fraction motion estimation (FME) part by skipping them when specific criteria are met inside a fraction execution resolver (FER) function. Our encoder already has a GPU-friendly fast integer motion estimation algorithm [10], so this extension focuses only on the fraction optimization part. As we target mostly the end users, we only exploit and capitalize common hardware setups such as video cards. Video cards with powerful graphics processor units

(GPUs) embedded in them cannot be considered an extra addition to these common setups; so, as nowadays GPUs are a standard hardware component for every video card unit, they cannot be considered as external hardware add-ons such as FPGAs, which have custom design requirements or costs for the end user, but rather as already built-in components. GPUs can be used for fast general-purpose calculations, so many researchers have exploited their capabilities. GPUs are bandwidth optimized but have low latency—unlike CPUs—so massive parallelism is used to hide latency. Although these programming considerations have been mentioned before, GPUs are continuously evolving to deliver new capabilities to all workloads even though they were initially intended for gaming.

One of the key technologies that has appeared recently in the latest GPU generations is tensor processing units (TPUs), which comprise an application-specific integrated circuit (ASIC) that charges for artificial intelligence (AI) calculation acceleration [11,12]. AI workloads are specifically designed to handle computational demands using matrix processor technology. Using TPUs as matrix processors instead of general-purpose processors, we can overcome the memory access restrictions that slow down both GPUs and CPUs. So, the primary task for TPUs is matrix processing, which is a combination of multiply and accumulate operations. In our proposed work, neither a GPU nor TPUs can be applied for the FME part for many reasons. The most important reason for the GPU's inadequacy is the problematic parallelization of the searching point dependencies (Figure 1), while GPU parallelization is a constraint-based scenario for sequential vectorized data.

Alternatively, as a suggested solution to this problem, a new GPU-friendly FME algorithm could be introduced to overcome the dependencies problem; however, this suggestion extends our proposed FER rather than mutually excludes it. Furthermore, TPU utilization was also dropped as an alternative acceleration proposition because the default FME algorithm generally requires high-precision arithmetic calculations to which TPUs are not suited [13]. In addition, since TPUs are designed for AI applications, they do not fetch instructions to execute directly; instead, the host loads the matrix processor for specific operations only [14,15]. So, currently, TPUs are inappropriate for general-purpose calculations, but, in the near future, TPUs can be developed in a such way as to outperform non-AI applications also. So, for the sake of (a) easy adaptation, i.e., no extra hardware is needed, and (b) easy implementation, i.e., intervention only at the software level with a well-known TSZ algorithm, we focused on skipping the encoder's default fraction ME when specific criteria are met.

We extended our previous hybrid encoding model [9,10] to support the proposed fraction execution resolver (FER) algorithm, which has been proven to be fast, easy to implement, and cost effective, minimizing in this way the computational load and thus the encoding times even further. This encoding model effectively utilizes both CPUs and GPUs in a flexible tiling selection scheme to balance speed and quality. Under specific circumstances where a balance between quality and speed is important for the end user, it provides a valid trade-off between speedup and video coding efficiency. We implemented a highly parallel environment using an innovative technique named Wavefront per Tile Parallelism (WTP). With this technique, we combined two parallelization schemes into one, maximizing in this way the parallelization workload of the encoder by the co-existence of WPP and tiles simultaneously (Figure 2), which is forbidden by design in HEVC and VVC specification standards. The primary encoding scheme of WTP is tiles, which are fully compatible with the mentioned standards. Furthermore, by extending our modified encoder, we enabled internally the Wavefront parallelization pattern for each tile separately, applying to the latter our multithreading environment setup. We propagated also the original structures to comply with the HEVC video standard, i.e., the encoder we selected to run our experiments [9]. Following the mentioned video standard, tiles divide each frame into independent rectangular regions. In our case, all tile columns and row boundaries are distributed uniformly, i.e., they all have the same size, and, essentially, they are treated as low-resolution separate frames—in CTU metrics—and they are encoded separately in a parallel pattern. So, each tile uses the standard WPP encoding pattern.

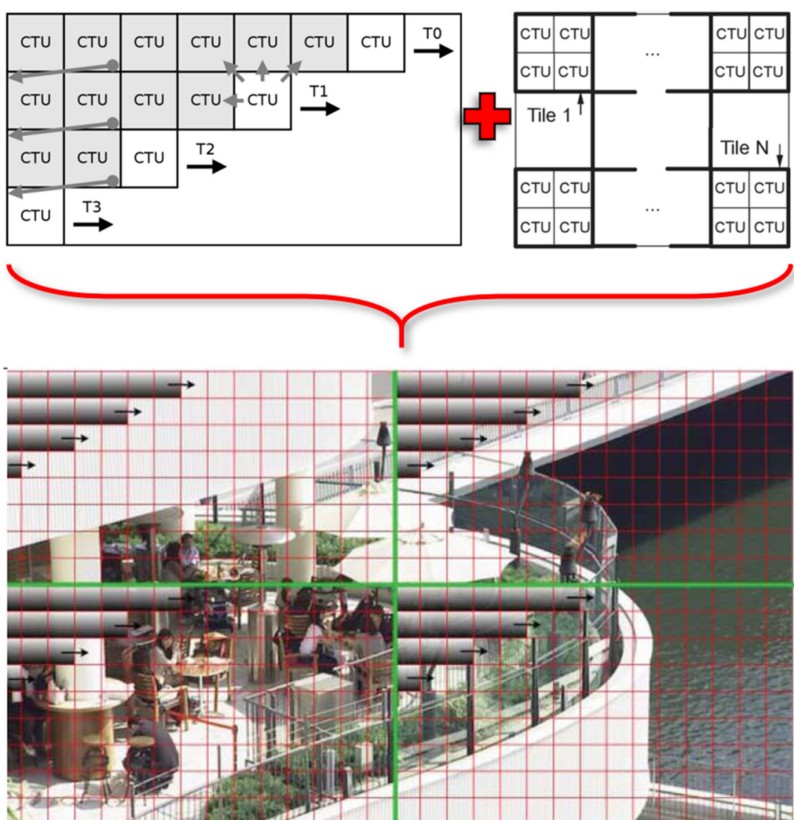

**Figure 2.** Combining WPP (**upper left**) and tiles (**upper right**) to create a new parallel scheme (WTP) (**bottom**).

GPU power is engaged [10] to improve the coding efficiency for each tile separately only for the integer motion estimation (IME) part and falls back to the CPU for the fraction refinement as a subsequent and mandatory job. The fraction part always follows the IME part, but it is very difficult to parallelize the default algorithm because of the dependencies it has. Vectorization is the particular way parallelism is achieved in this case. In particular, vectorization mostly uses dedicated SIMD execution hardware units in processors using specialized instructions such as ×86 SSE/AVX, ARM Neon, and GPU compute units. So, since we cannot achieve vectorized data in the FME case and the data size is not big enough for bulk calculations (non-bandwidth optimized), dedicated hardware may be the solution for this particular problem, which is costly, or a GPU-friendly FME algorithm that directly follows the IME calculations in the same kernel. Although there are parallelization implementation difficulties in the FME part, the fractional refinement is very important because it can increase the accuracy of the IME step using an interpolation pattern for the half and quarter samples of the testing area. While our high parallel CPU/GPU encoding scheme algorithm reduces significantly the encoding time, the fraction part (FME) that is always executed as a subsequent job does not contribute to the final encoding time since it is not optimized in any way.

Common video sequences are known that contain many static blocks or non-moving parts in each frame. In some special cases, there are even a number of whole consecutive frames that are static. Based on this idea, we optimized our encoder to invoke the fraction part if, and only if, there are non-static blocks inside the current frame and skip it completely when this is not the case, which means that the fraction part is ignored if static blocks are met. We implemented an algorithm that applies or skips the FME if specific criteria are met. The fraction execution resolver (FER) is a software solution algorithm that minimizes the encoding time by avoiding unnecessary calculations. In the same multithreaded environment, with an identical setup, our proposed FER algorithm can achieve a greater

than 42% encoding time saving when enabled at its peak, i.e., a sequence of still images (150 frames with a repeated photo for all frames) (Figure 3). It is worth mentioning that the PSNR and bitrate remain unchanged, i.e., there is no loss or image degradation compared to when the counterpart with the same setup but without FER is used. Regarding the total encoding time saving when compared to the setup without GPU-IME acceleration and FER disabled (original WTP), the gain is more than 52%, and, finally, compared with the setup with the default (single-threaded) original HEVC encoder, the gain is more than 2285%. These experimental results were the best we could obtain for the specific hardware setup in the identical case where all frames repeat the same stationary image. With common video sequences, the gain is more than 30%, since most of them have more or fewer moving blocks. So, our proposed work focused exclusively on the fraction execution resolver (FER) algorithm and the criteria that should be met to decide when to execute or discard the fraction part. The main criteria behind this decision are (a) zero motion vectors (MVs) for both XY vectors at the IME part and (b) zero distance displacement from the predicted search area center. This test is executed only once for the initial $64 \times 64$ ($2N \times 2N$) block. For that purpose, we created a preliminary fast test decision point (pFTDP) function which is a modified, cut-down version of the original diamond test zone search (TZ) algorithm [16].

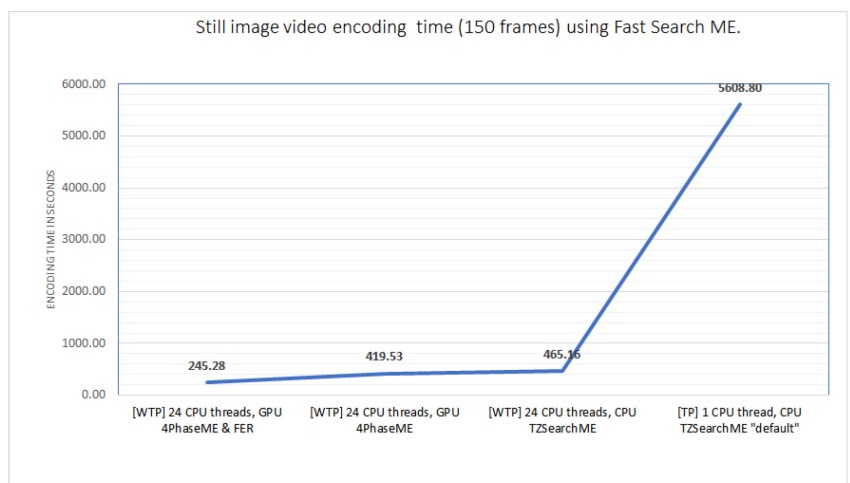

**Figure 3.** Encoding time with different setups. Fraction execution resolver (FER) achieves 42% encoding time savings compared to the same optimized setup without the FER extension ([WTP] 24 CPU threads, GPU 4PhaseME). Where [WTP] is used, bitrate and PSNR were retained unchanged.

## 2. Related Work

Many solutions have been presented in the past to accelerate FME computations and motion estimation in general. They are based on two categories: (a) software-optimized algorithms and (b) hardware implementations. Hardware implementations include non-CPU integrated circuits such as FPGAs (field-programmable gate arrays), ASICs (application-specific integrated circuits), and video card GPUs. The video card is an exception because it is compulsory for the system functionality itself. Thus, it should be treated as a de facto piece of hardware installed in the system, like CPUs, RAM, or any other important component, and not as extra hardware with a custom design or requirements.

A high-throughput hardware architecture for the FME interpolator was proposed [17,18] using specialized hardware that always uses the FME part but in an optimized way. The results showed that a design architecture targeting an Altera Stratix III FPGA device is capable of processing real-time QFHD ($3840 \times 2160$) resolutions at 60 fps with a 353.8 Mhz clock frequency and 995 Mpixels/s for $7680 \times 4320$@30 fps at 188 MHz on a 64 nm VLSI CMOS chip. To the best of our knowledge, most hardware implementations focused only on FME optimization algorithms rather than skipping that part. An exception to this rule, with a similar approach to our basic skipping concept, was implemented on an FPGA hardware platform by F. H. Shajin [19] using a quadrant-based search algorithm. A zero-movement

prejudgment technique (ZMP) is used to find out whether a block is static or not. This decision is used to decrease the computational complexity of the search method since ZMP can identify stationary blocks (distortion of the block is under threshold (T) forecast) even before the search procedure begins. The latter method uses a custom search algorithm (quadrant based) that is tailored to work on a Verilog HDL with Virtex-5 technology and integrated with the Xilinx ISE Design Suite 14.5 hardware platform. It achieves lower search points compared with the hexagon, adaptive root pattern, and diamond search algorithms. Using lower search points, the quadrant-based search algorithm provides a PSNR of 10.906%, 2.03%, and 8.3562% and motion estimation time of 1.89%, 1.4693%, and 0.131% compared with the aforementioned algorithms.

Approximate unified FME filters (AUFF) are another hardware implementation [20] for interpolation filters that use two hardware designs. Using a 40 nm standard-cell library, with a power dissipation ranging from 22.04 to 62.06 mW, the hardware is capable of real-time interpolation of UHD videos of 2160p@60 fps and 4320p@60 fps when synthesized. Gogoi et al. [21,22], using special hardware in HM 16.8, achieved an 8.725% and 9.072% time saving using a low-complexity algorithm with 38 search points in two different pattern structures (PS). It provides real-time encoding of $4096 \times 2160$@60 fps operated at the maximum frequency of 353 MHz. Although all mentioned solutions reduce the computational complexity, affecting the coding efficiency more or less significantly, undoubtedly they are specialized hardware implementations that do not focus on the end users but rather target companies and professionals that need real-time live streaming or broadcasting encoders. Much hardware equipment that everyone has broad access to, such as modern superpower GPGPUs (general-purpose graphics processing units), comes with next-generation, built-in encoding-dedicated hardware such as Nvidia's NVENC, AMD's AMF, and Intel's Quick Sync. They offer real-time streaming characteristics but with predefined codecs (i.e., h.264 or AV1) and image quality presets. No custom optimization tampering is possible since the encoder is closely connected to the hardware.

Software solutions are easy to implement, have low costs, do not need extra hardware, and mostly target minimization of the computational complexity for low-end users. Fast two-step sub-pixel motion estimation is proposed in [8] which can reduce the sub-pixel search points significantly with negligible quality degradation. This algorithm uses six-pixel integer points for the fraction part. The half pixel is obtained from a sub-pixel error model surface that is first applied. Using the information resulting from the latter step and an additional integer point, a new error model surface is applied in a smaller area. Thus, the quarter-pixel point is obtained by minimizing the function. The best MV candidate is the one with the minimum RD cost among integer, half-pixel, and quarter-pixel points. For the standard Class B ParkScene video sequence, the encoding time saving is 46.25% with a −0.02 PSNR quality drop compared with the hierarchical search (HS) algorithm.

In [6], an adaptive fractional pixel ME skipped scheme was proposed for low-complexity ME. In this scheme, all children's PU modes are classified into their root-type PU ($2N \times 2N$) during the encoding. Based on the ME earning result of the root type, the FPME of all sub-pattern PU modes is skipped. The proposed algorithm reduces the encoding complexity and maintains a comparable encoding efficiency with the original HM video standard. Even though the basic idea is similar to our proposed work, the criteria to skip the FPME part and the computational load before that decision are very different. The root-type PU mode (Inter_2Nx2N) performs both integer and fraction ME parts. If the best motion vector of both IME and total ME (IME + FME) is equal, then the FME can be skipped. The fast search algorithm used in the IME part in HEVC is the test zone search (TZS), which, by default, has three different searching modes [16]: (a) the zonal search, (b) the raster search, and (c) the refinement. If the zonal search fails (best MV candidates are far away from the center of the search area), then a raster search is applied, which is essentially a full search in a down-sampled search area, a process that slows down the ME efficiency. Thousands or even millions of Inter_2Nx2N CTUs may exist in a video sequence to which a raster search may be applied, minimizing in this way the coding efficiency.

In our proposed work, the pFTDP function is a lightweight zonal search only. This means no raster search is performed by default; thus, a huge time saving can be achieved for the encoder itself. Another major difference is that our proposed pFTDP does not even need to execute the fraction motion estimation part for the root-type PU mode to make its decision, which also adds extra computational overhead. Our skipping criteria are decided inside a lightweight preliminary function that calculates at the same time both the best MVs and minimum RD cost for all static blocks. Another adaptive fraction motion skipping process was proposed by Lee et al. [7] for multiview extension (video data captured from different viewpoints) of the standard HEVC encoder, called multiview HEVC (MV-HEVC). The decision-skipping condition, in this case, takes into account the interview frame encoding that performs a disparity estimation (DE) to reduce redundancies between neighboring views. In this case, the algorithm is designed to reduce the computational complexity of both ME and DE. When the result of 2N × 2X PU is $Inter_{2N}^{T} - I$, the faction and disparity part may be skipped. If the result is $Inter_{2N}^{IV} - I$, then only the fraction part is skipped. This FME skipping method uses a different video coding structure that performs the inter- and interview coding. The encoding time saving is 27.71% with a coding loss of 0.07% on average.

A parallel pre-motion estimation with IME and FME was proposed by Zhang et al. [23]. The efficiency of data access is achieved using a rapid mapping table algorithm. The GPU power is used to calculate SAD or SATD efficiently for blocks of different sizes for both IME and FME parts. A maximum of ×12.13 is the average speedup for IME and FME parts—both implemented in the GPU kernel—for the Class B sequence Basketballdrive (the only video sequence that can be compared with our experiments), which is much lower than our ×15.54 speedup experiment result without the need to implement a GPU-dependent FME algorithm. Finally, another GPU-based parallel algorithm was proposed by Luo et al. [24]. They optimized the whole ME process by separately optimizing hierarchically the CTUs (coding tree units), the PUs (prediction units), and, finally, the MVs (motion vectors). The latter proposed scheme accelerates the motion estimation by up to 12.7 times with a BD-BR loss of 0.52% on average. Other implementations [25–27] made use of the GPU parallelization power either using only the IME part or the FME part, but many concessions were made to achieve the best parallelization results. All of the referenced works tried to reduce the FME computational complexity, either by using specialized hardware or by applying an optimized FME algorithm. So, our efforts in this work were focused mostly on software solutions, emphasizing the popular TZS algorithm, which is considered to be one of the fastest ME algorithms [28]. Our proposed fraction execution resolver approach differentiates mainly in three ways: (a) it does not require any special hardware equipment; (b) it is super fast since CPU cores are latency optimized for a lightweight job, i.e., execution of the pFTDP function; and (c) it is easy to implement as the pFTDP decision-maker function is just a modified version of the well-known TZS algorithm. To sum up, we propose a software decision-maker algorithm to decide the invocation of the FME part based on the encoding process without any significant losses. The negative effects of a maximized tiling pattern concern the WTP encoding scheme itself more than the FER algorithm. These effects can be improved when a lower tiling pattern is selected. With super-high-resolution video sequences, the gain of the WTP/FER was expected to be higher even with lower tiling scheme patterns since the tile's internal parallel Wavefront progressively engages more CPU cores than low-resolution videos. Finally, our FER approach can be extended to include a multicore GPU/TPU-friendly fraction estimation algorithm or even a GPU-kernel-based pFTDP function to further improve the coding efficiency. Emerging deep learning methods can also be applied, such as implicit functions and attention mechanisms, to predict the motion vectors from the past GOP (group of pictures) and thus speed up the encoding process. Nevertheless, such extensions exceed the scope of this paper, which proposes a fraction execution resolver and evaluates its potential merits, leaving further optimizations for future work.

## 3. Implementation

To evaluate our proposed algorithm, we used our proven multithreaded Wavefront per Tile Parallelism (WTP) encoding scheme [9]. This modified encoder uses tiles as a primary parallelism scheme but internally uses WPP for each tile separately with multiple CPU cores working in a parallel pattern. Extending the original WTP encoder, the GPU's fast search algorithm has been demonstrated (4PhaseFS) [10] to speed up the motion estimation calculations, which has been proven to be one of the most intensive tasks in a video encoder. As in our previous works, to achieve the best performance from our experiments, the application thread pool was fed with a pre-selected number of threads that matched exactly with the physical CPU cores we had available for our setup. With respect to the video sequence's available resolution, the speedup automatically scales [29] depending on the CPU and GPU capabilities [30]. The GPU contributes significantly to the minimization of the motion estimation computation cost [31–35] since many ME vectorized data can take advantage of this powerful piece of hardware. The OpenCL framework [36] was the ideal candidate to set up the GPU kernel and schedule its cores to compute the integer estimation part (IME) in a parallel pattern. OpenCL is a universal general-purpose graphics processing unit programming framework that major GPU manufacturers support; it has an easy learning curve, and it seems to have higher performance rates in comparison with other competitive frameworks [37]. In the WTP parallelization scheme, each CPU thread executes its own copy of the code in a sequential pattern. When a thread meets the beginning of the motion estimation's IME part [38], it automatically reverts to the GPU kernel, trying in this way to distribute the encoding resource load among different hardware parts of the system. Even though the thread waits for the GPU to send back its calculation results, the OS frees some of the thread's resources, allowing in this way the normalization of the load balance of the encoder. The GPU exceeds the vectorized data calculations in comparison with the CPU because it encapsulates many thousands of cores that work in parallel. The GPU kernel program (a C-like program) uses a parallel reduction method [39] to calculate the distortion ratios. A bottom-up accumulative pattern is used, starting from $4 \times 4$ blocks as a base for the next SAD (sum of absolute differences) calculations, and progressively builds bigger PU blocks until it reaches a maximum of a $64 \times 64$ root-type PU. An OpenCL work group of $16 \times 16$ is selected (256 cores in total) to comply with the initial $4 \times 4$ blocks, which consist of 256 pixels. Next, in the SAD calculation kernel steps, each core may apply to more than one-pixel calculation offset to optimize the GPU resources. This means that each CPU thread uses a set of 256 GPU cores at the current time, so a total of 6.144 GPU cores may be needed for 24 CPU threads if all of them concurrently request resources from the GPU. This is an extreme possibility, but it is possible in UHD video sequences. In such a case, the GPU driver sets the extra demands in a wait queue. Finally, the kernel (GPU) returns to the caller (CPU) a set of indexed arrays—concerning the whole search area [40]—with the minimum distortion ratios as well as the corresponding motion vector. To minimize the data transfer costs, memory optimization strategies are used between the kernel and the host [41,42].

The GPU kernel, as described above, takes into account only the integer part (IME), while the fraction part (FME) is left to the CPU since the parallelism of the fraction part inside the GPU is inefficient due to its dependencies. The data transfer between the CPU and GPU and vice versa is costly, and it is worthwhile only if heavy calculations in parallel are to be made. For light calculations such as the one made in the pFTDP function, the CPU is faster, without slowing down the whole process sending the preliminary calculation directly to GPU; thus, costly data transfers are avoided. Based on this idea, we developed a preliminary fast test decision point (pFTDP) using a modified but optimized version of the original TZSearch. The pFTDP function takes into account only the first part (zonal search) of the original TZ searching algorithm with a maximum distance of two (2), i.e., checks only neighbor areas of the $64 \times 64$ block with distances of 1 and 2 according to the predicted starting position. After the end of this preliminary stage, if the MVs are still zero and the best distance is also zero, we can safely conclude that the current block is a static block. In

this case, we can use the already calculated motion vectors and RD from this lightweight function to return the results back to the caller, discarding the fraction part. Otherwise, i.e., we have a non-static block, we call the GPU kernel to precalculate MVs and RDs for all sub-blocks [10] at once and use them as an input parameter for the fraction part (Figure 4) in the next stage. The decision making is executed only for the initial 2N × 2N of each block pattern, while the subsequent block patterns follow the flow taken from this initial decision, i.e., they either (a) obtain the MVs and RD from the lightweight pFTDP function while FME calculations are skipped (static blocks), or (b) use the precalculated GPU MVs and RD in case the pFTDP fails, and, consequently, FME calculations are executed for each one of them (non-static blocks) from the CPU. So, if the pFTDP succeeds, i.e., we have a static block, then there is no need to call the heavier GPU kernel since the gain would be negligible if not zero, and FME calculations are also bypassed since there is no need to obtain more accuracy from the specific static CTU block. The already calculated MVs and RD from this stage are sent directly to the caller function since they are the best candidates we can obtain so far, and the GPU is not involved. The extra optimized method applied here—as already mentioned—is the one-time execution of the pFTDP, i.e., when a 2N × 2N (64 × 64) block pattern is met.

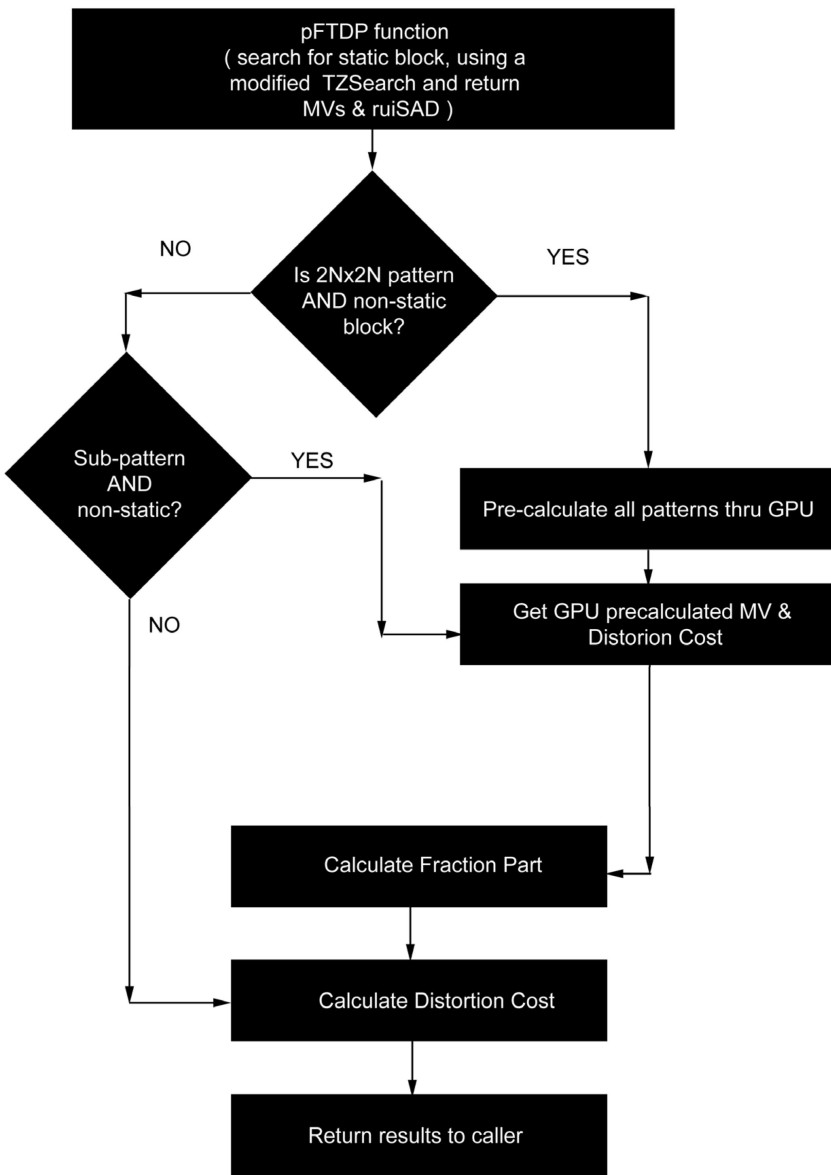

**Figure 4.** The fraction execution resolver (FER) flow chart.

According to the strategy described above, it is obvious that we are using a hybrid computation encoding scheme pattern for the motion estimation part; (a) non-static integer CTU blocks are calculated in parallel using the GPU kernel and FME from the CPU, and (b) static CTU blocks, also in parallel but exclusively from the CPU, at the same time skip the FME part. Internally, our pFTDP algorithm still uses the eight-point diamond search around the starting point, but the only distances compared across the reference search area are those of step 1 and step 2. The pFTDP function always returns the evaluated results of *BestX*, *BestY*, and *BestDistance*. If the block is 2N × 2N, we immediately sign our code from the very beginning, either as a static or a non-static block, and, consequently, the fraction execution (FME) is invoked. This is valid also with all subsequent block patterns that follow the initial sign of the pFTDP and consequently execute or skip the FME calculations.

Lines 1–9 of Figure 5 are the initialization part of the algorithm. Lines 10–21 are the main loop where the TZ eight-point diamond search is applied. If the points around the center return zero for both X and Y motion vectors and the distance is also zero, we break the loop with true for the flag isStillBlock, which is used by the caller as a decision point for further processing (Figure 3). If the distance is higher than 2, then the isStillBlock flag returns to the caller with false. Lines 22–23 calculate the ruiCost and return back to the caller the memory address of the BestX and BestY candidates as well as the isStillBlock decision flag.

```
Input   : SearchRange passed from xMotionEstimation function
Output  : IsStillBlock, BestX, BestY, ruiCost
Notes   : ptr denotes memory address of X variable. It is used for both input and output.

 1:function pFTDP(ptr SearchRange, ptr iBestX, ptr iBestY, ptr ruiCost)
 2:  IsStillBlock ← false;
 3:  iBestX = iBestY ← 0;
 4:  call function getPredictedXY(ptr iBestX, ptr iBestY)
 5:  iStartX ← iBestX
 6:  iStartY ← iBestY
 7:  iDist ← 0
 8:  uiBestRound ← 0
 9:  uiBestSAD ← 0
10: while iDist < SearchRange
11:        call function xTZ8PointDiamondSearch(ptr iStartX, ptr iStartY, in iDist, ptr iBestX, ptr iBestY, ptr uiBestSAD, ptr uiBestRound)
12:        if  uiBestRound >= 3
13:        begin
14:          if uiBestDistance = 0  AND iBestX = 0 AND iBestY = 0
15:          begin
16:              IsStillBlock ← true;
17:          end if
18:          exit from loop
19:        end if
20:        iDist ← iDist mul 2
21: end while
22: ruiCost ← call function calculateCost(ptr iBestX, ptr iBestY, ptr uiBestSAD)
23: return IsStillBlock
24:end function
```

**Figure 5.** The preliminary fast test decision point (pFTDP) algorithm.

## 4. Experiments

### 4.1. Implementation Details

Our code was implemented in HM 16.17 reference software. The programming language used to optimize the encoder was C++. Furthermore, the AMP (asymmetric motion partitioning) inside the configuration file was enabled to increase the accuracy of the motion estimation part when it is needed and, consequently, increase the computational load for comparison reasons between CPU and GPU setups. Finally, the thread pool was configured with 24 CPU threads to comply with the physical CPU cores of our setup. We kept the CPU thread assignment strategy and routing used in our previous work [10] for comparison reasons, meaning that, if any CTU encoding dependency constraints are met, the active CPU thread is dismissed immediately back to the threading pool to avoid wait state cycles and to free up unused CPU/GPU resources.

*4.2. Setup*

Experiments were conducted on a two 12-core 2.20 Ghz Intel Xeon E5-2650 Linux server for base frequency, with 2.90 GHz for turbo mode and 1.20 GHz for power-save mode. The GPU computations for the integer motion estimation part were made with a PCI Express 3.0 × 16 Nvidia Quadro P4000 video card. Quadro P4000 was introduced in April 2016 by Nvidia, implementing at the time the new Pascal architecture that allows multi-kernel execution as independent units, and it combines 1792 CUDA cores with 8 GB of memory.

As shown in Table 1, common Class B and A video sequences [43] were used in our experiments. Due to the number of computationally intensive tasks and time restrictions, we used the first 150 frames of each video sequence to evaluate the proposed FER algorithm. A GOP size of 4 was used with a low-delay (LD) setup, meaning an <I> frame was initially used, followed by consecutive <B> frames. The following parameters were used unless otherwise stated: 64 × 64 CTU size, TZ fast search mode, one slice, max partitioning depth of 4, 24 CPU threads, and default QP = 32. Tile partitions 2 × 2, 3 × 3, and 6 × 4 remained the same as in our previous work [10] for comparison reasons.

**Table 1.** Test video sequences.

| Class | Sequence Name | Resolution |
|:-----:|:-------------:|:----------:|
| B | Basketballdrive | 1920 × 1080 50 fps |
| B | Bqterrace | 1920 × 1080 60 fps |
| B | Cactus | 1920 × 1080 50 fps |
| B | Kimono | 1920 × 1080 24 fps |
| B | Parkscene | 1920 × 1080 24 fps |
| A | Peopleonstreet | 2560 × 1600 30 fps |
| A | Traffic | 2560 × 1600 30 fps |

*4.3. Coding Efficiency Results*

In most of our experiments, we measured the speedup using the optimized WTP (GPU-IME accelerated) with FER over the default TZ search (one thread, sequential) for the same settings with different QPs but the same number of threads. The average speedups for Class A and B sequences are shown respectively in Figures 6 and 7. In both plots, the average rate speedups are more than ×12 for Class B videos and more than ×14 for Class A videos. In Figure 8, it can be observed that the BD-Rate is proportional to the tiling scheme used, i.e., a partitioning scheme with a smaller number of tiles has lower quality loss and vice versa. This was expected because tiles reset the context variables of the CABAC (context-based adaptive binary arithmetic coding) at their initialization step. This means that the CABAC learning process starts over again with zero coding units being used from the neighboring prediction encoding history. Thus, a PSNR decrease and BD-Rate increase are expected. This means that a bigger tile pattern scheme, i.e., 6 × 4, capitalizes the CPU usage in our setup (all 24 CPU physical cores are involved), but, at the same time, results in a relatively higher BD-Rate. The same BD-PSNR [44] decrease effect can also be observed in Figure 9, where a −0.22 quality drop is plotted for the PeopleOnStreet video sequence. For this plot, the summed average PSNR values for all QPs (= 22, 27, 32, 37) were used to calculate and plot the final BD-PSNR. In Figure 10, we show the absolute speedup increase compared with the default HEVC encoder benchmarks results when FER is enabled in our GPU-IME accelerated WTP [10] and without this. Finally, the last plots (Figures 11 and 12) show the PSNR (image quality) and bitrate overhead when FER is enabled in our WTP (GPU-IME accelerated) encoder. The PSNR appears to be slightly improved (positive values in Figure 11), while, in other cases (negative values), the image quality drops but is proven to be negligible; −0.05 in the worst-case scenario. The explanation for the slight image quality improvement is that the pFTDP algorithm is more accurate than the GPU-IME 4Phase fast search counterpart

in some cases, i.e., in static blocks. The bitrate overhead shown in Figure 12 seems to be almost consistent in both experiments (FER enabled/disabled). The slight improvements or deteriorations observed are more related to the IME algorithms themselves, i.e., CPU-based pFTDP versus GPU-based 4PhaseME counterpart.

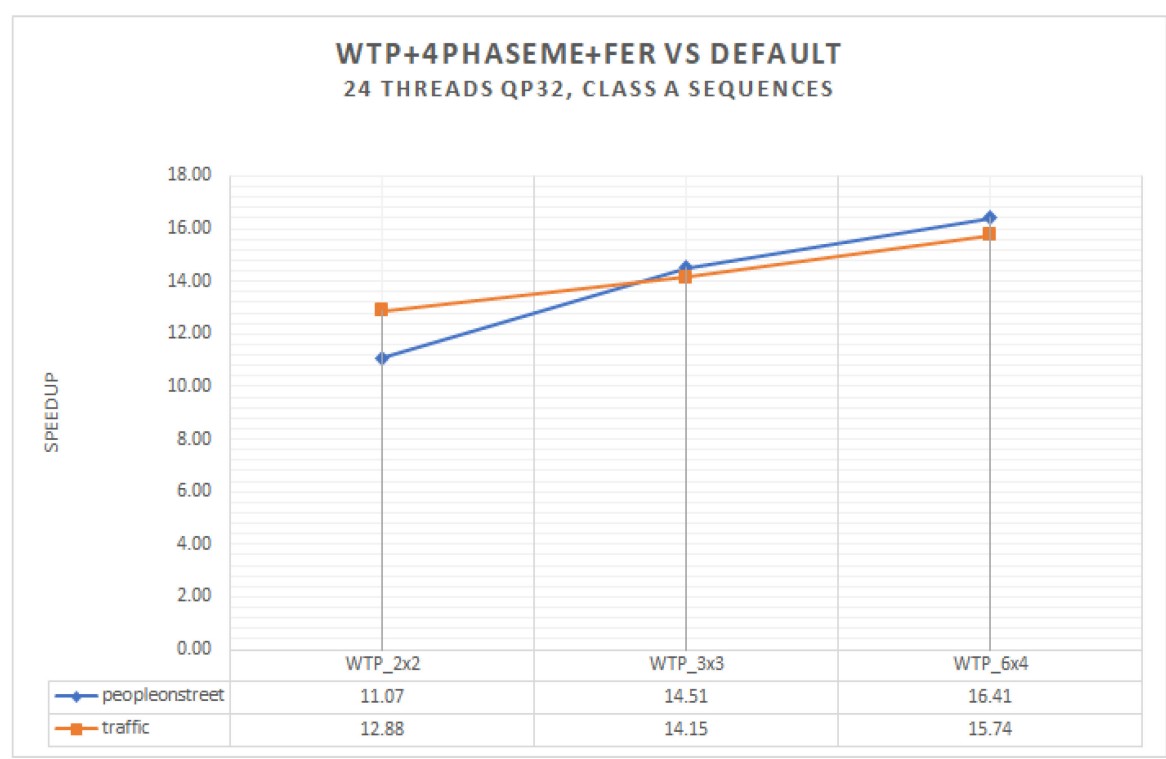

**Figure 6.** Class A sequences using FER with WTP speedup for QP = 32 and 24 threads versus default.

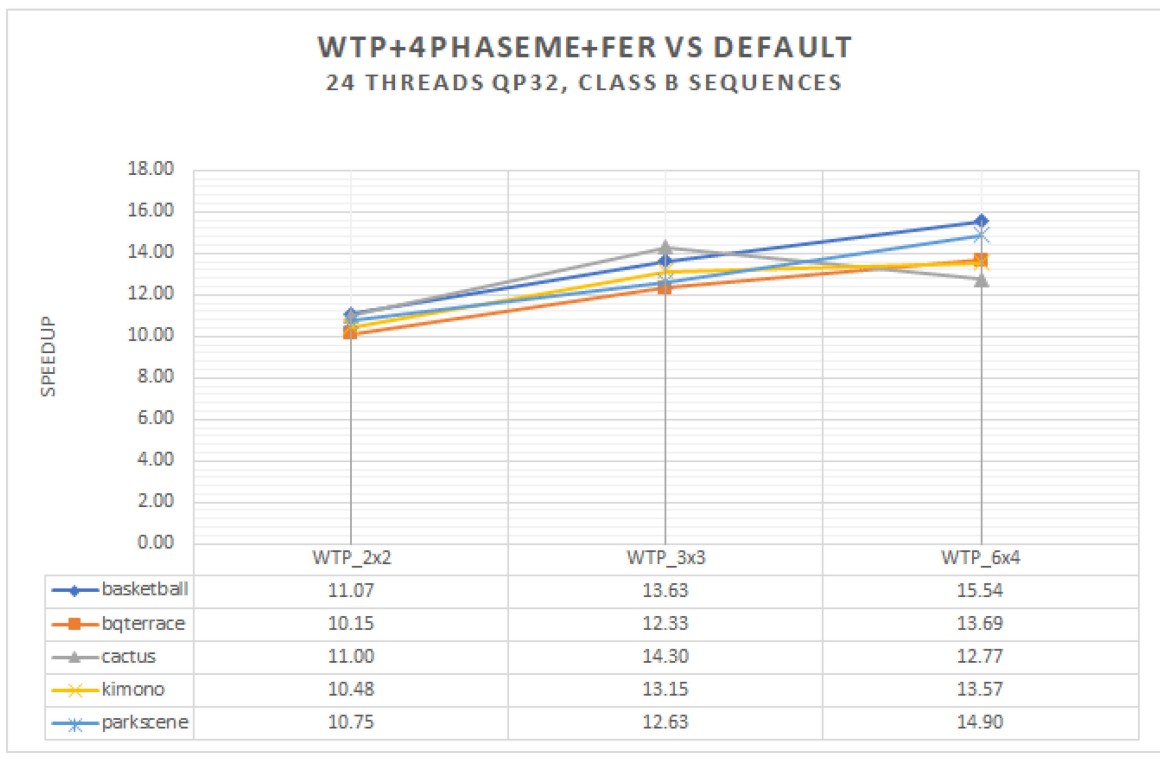

**Figure 7.** Class B sequences using FER with WTP speedup for QP = 32 and 24 threads versus default.

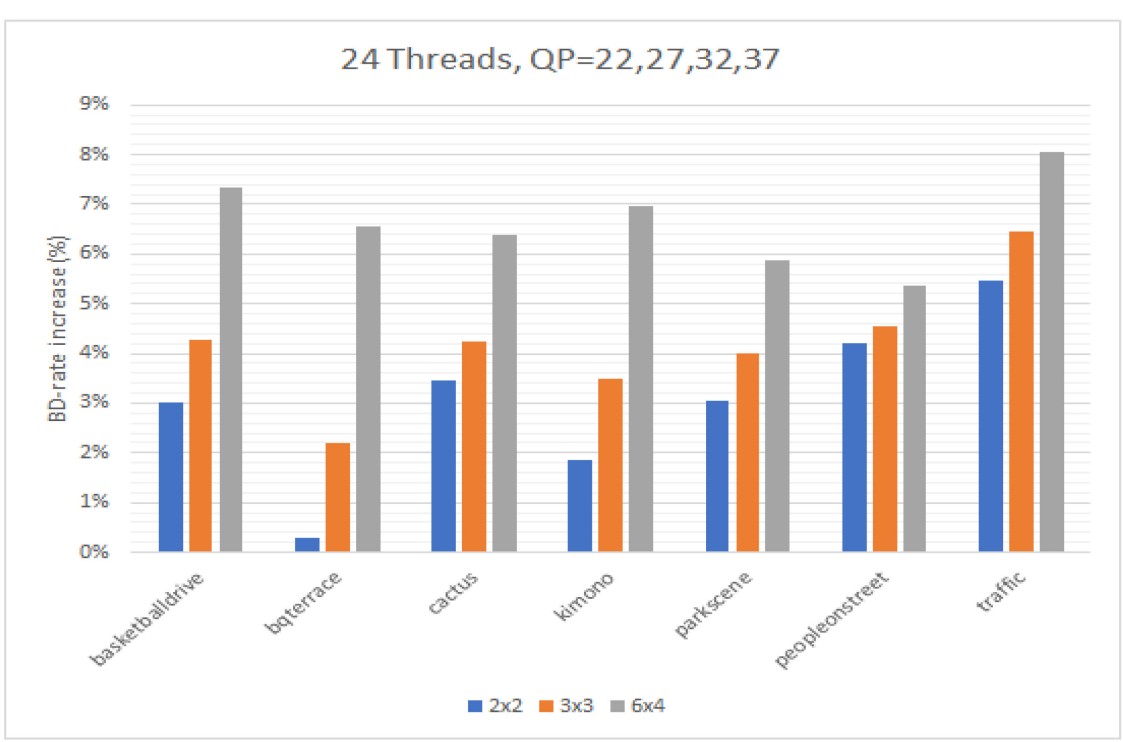

**Figure 8.** Increase of BD−Rate using FER with WTP for QP = 22, 27, 32, 37, and 24 threads versus default (positive values indicate bigger files).

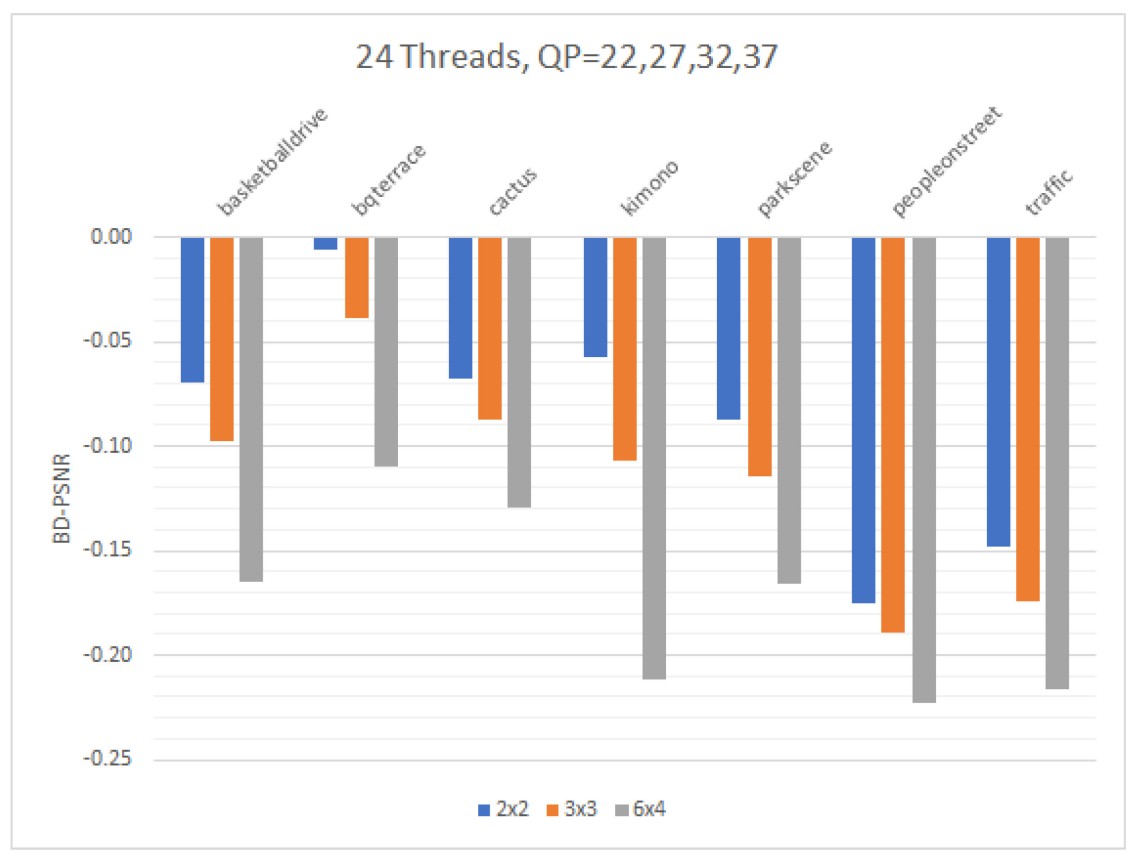

**Figure 9.** Decrease of BD−PSNR using FER with WTP for QP = 22, 27, 32, 37, and 24 threads versus default (negative values indicate a quality drop).

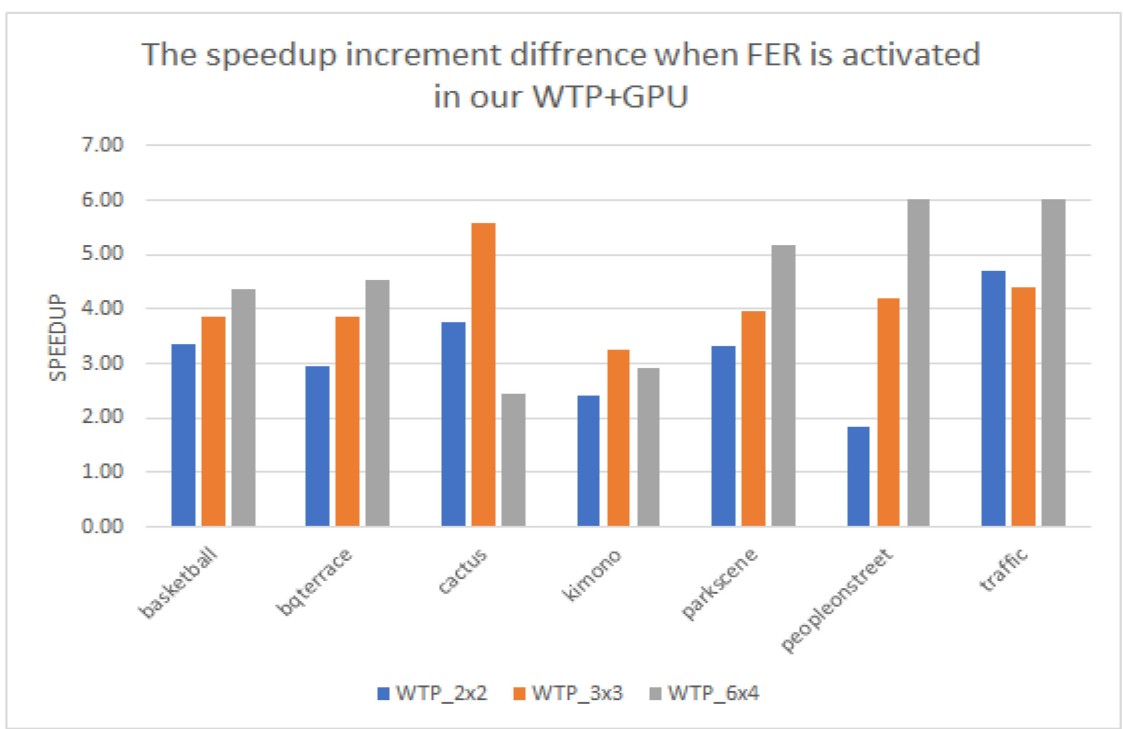

**Figure 10.** The absolute speedup increments when FER is enabled/disabled in our GPU-IME accelerated WTP encoder compared with the default HEVC encoder benchmark results, i.e., Diff (FER enabled–FER disabled).

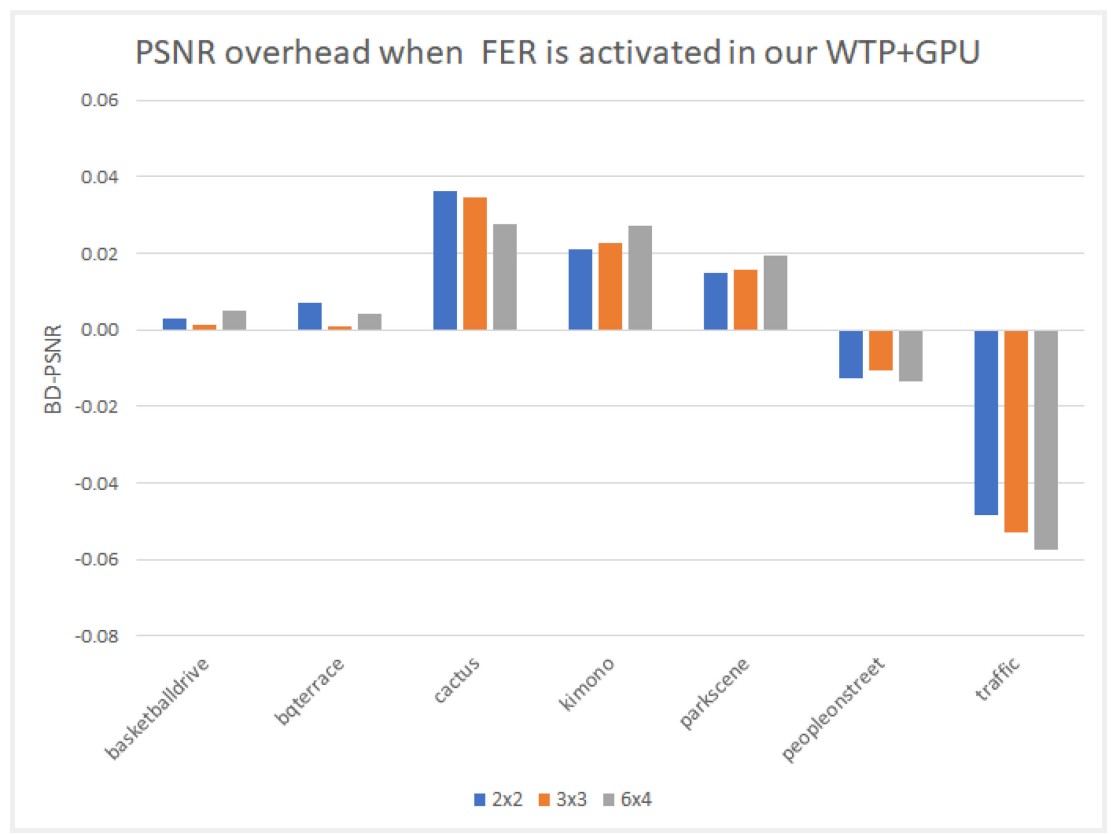

**Figure 11.** The PSNR overhead when FER is enabled in our GPU−IME accelerated WTP encoder. Positive values indicate a quality improvement, while negative values indicate a quality drop.

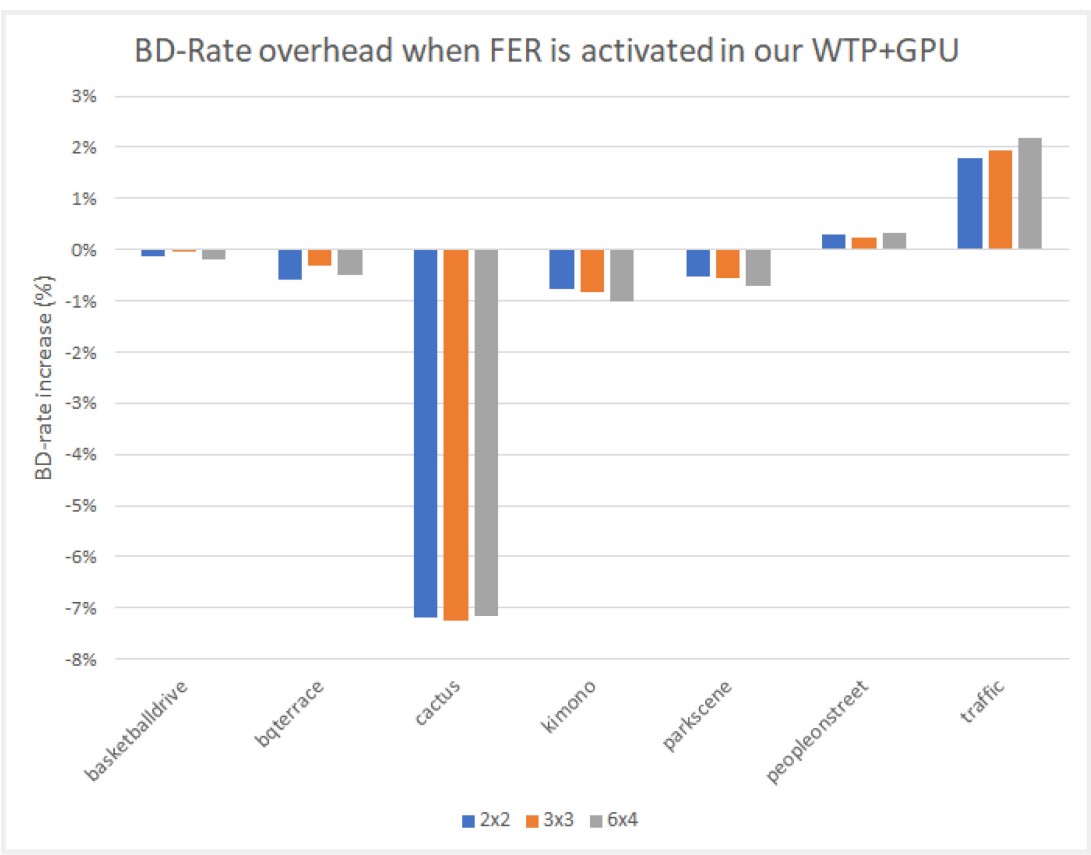

**Figure 12.** The bitrate overhead when FER is enabled in our GPU−IME accelerated WTP encoder. Positive values indicate bigger files.

## 5. Conclusions

In this paper, we presented a fraction execution resolver (FER) algorithm that works in parallel and non-parallel encoders and aims to minimize the encoding time and overcome the standalone deficiencies of complex computations and specialized hardware. Results demonstrated that the FER can achieve an average speedup of ×14.3 for Class A video sequences and ×12.6 for Class B video sequences with an average PSNR decrease for both Class A and B video sequences of −0.12 and an approximate 4.6% increase for the bitrate. The average speedup enabled by the FER, in comparison with the same sequences and identical setup without the FER, is about ×3.95. Video sequences that have many static blocks gain higher time-saving encoding times using the proposed FER algorithm. It is worth highlighting that the FER concept can easily be adapted with many other video encoding standards, such as newer VCC, since it is not hardware dependent, and similar decision criteria can be applied. According to the results demonstrated, a PSNR and bitrate overhead occurred. These negative effects are more related to the tile encoding scheme itself and the way CABAC works internally. The FER itself has been proven to add negligible overhead in PSNR and bitrate metrics. To combine speed and quality, an encoding balance scheme should be used via a lower tiling pattern, preferably a $2 \times 2$ scheme, to obtain the best of both worlds. Cumulatively, the experimental results confirmed the validity of our motivation, namely, that we can benefit from a software fraction execution resolver without any extra hardware costs or considerable quality loss. The gain is further increased when video sequences used for encoding have more static blocks than others; however, practically, static blocks always exist, more or less, in common video material.

**Author Contributions:** Conceptualization, G.I.P.; methodology, G.I.P. and T.L.; software, G.I.P.; validation, G.I.P., M.K., T.L. and I.A.; writing—original draft preparation, G.I.P. and T.L.; writing—review and editing, G.I.P., M.K., T.L. and I.A.; supervision, T.L.; project administration, M.K. All authors have read and agreed to the published version of the manuscript.

**Funding:** This research received no external funding.

**Data Availability Statement:** Data available in a publicly accessible repository.

**Acknowledgments:** We acknowledge the support of this work by the project "ParICT\\_CENG: Enhancing ICT research infrastructure in Central Greece to enable processing of Big data from sensor stream, multimedia content, and complex mathematical modeling and simulations" (MIS 5047244), which is implemented under the action "Reinforcement of the Research and Innovation Infrastructure", funded by the operational program "Competitiveness, Entrepreneurship, and Innovation" (NSRF 2014-2020), and co-financed by Greece and the European Union (European Regional Development Fund).

**Conflicts of Interest:** The authors declare no conflict of interest.

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
