# Peer review of "Fraction Execution Resolver Using a Hybrid Multi-CPU/GPU Encoding Scheme"

_electronics, doi:10.3390/electronics12173586_

Round 1

Reviewer 1 Report

The propose has significantly improve the state of the art performance. However, my major concern is that the study need to consider field-programmable gate arrays (FPGAs), and  tensor processing units (TPUs) to prove the advantage of the propose hybrid CPU/GPU. The discussion should continue by adding the possible reasons why the propose scheme perform better, explain the implications of the study to theory and practice; lastly discuss the limitations of the study and suggest future research direction.

The paper is not up to date only one paper of 2021, no 2022 papers, no 2023 papers, this indicate that the study is not aware of the current developments in the research area. Rigorous LR should be conducted to show that what was propose in the study has not been done between 2021 to 2023

Just a few minor edit before final publications

Author Response

Response to Reviewer 1 Comments

Thank you for the valuable feedback and your comments.

Point 1: However, my major concern is that the study need to consider field-programmable gate arrays (FPGAs), and  tensor processing units (TPUs) to prove the advantage of the propose hybrid CPU/GPU. The discussion should continue by adding the possible reasons why the propose scheme perform better, explain the implications of the study to theory and practice; lastly discuss the limitations of the study and suggest future research direction.

Response 1:  We added an extra paragraph (lines 61-70) justifying why we dropped the FPGAs or TPUs, and the pros and cons of similar schemes. This extra paragraph also explains why this scheme performs better under specific circumstances which are cost flexibility and simplicity. Also, lines 334-340 unfold some thoughts about limitations and future research directions

  Point 2: The paper is not up to date only one paper of 2021, no 2022 papers, no 2023 papers, this indicate that the study is not aware of the current developments in the research area. Rigorous LR should be conducted to show that what was propose in the study has not been done between 2021 to 2023

Response 2: We focused our research on a very specific topic which is the Fraction Motion Estimation skipping algorithms. To the best of our knowledge, there are no updates on this topic. Although, after your suggestion, we opened our research to similar topics and updated our references.

Reviewer 2 Report

The idea is interesting, but there is limited support for all the claims in the paper.

Whether the gains carry over everywhere is an unanswered question, as goes with anything in simulations alone.

Author Response

Response to Reviewer 2 Comments

Thank you for the valuable feedback and your comments.

Point 1: Whether the gains carry over everywhere is an unanswered question, as goes with anything in simulations alone.

Response 1: The Test Zone Search (TZS) algorithm was adopted in both HEVC and VVC video standards. Also, many other hardware and software encoders are using TZS or a variant one for their motion estimation. Our pFTDP function is a cut-down modified TZS algorithm keeping the same functionality and variable naming. The pseudocode in Fig 5, shows exactly the modification code, and the criteria are used to skip the FME part. The experiment results are conducted with HM 16.17 reference software in real-world environments (Linux and Windows) using a C++ code and not in a simulated one like MatLab. Please refer to paragraph 4.1 Implementation details.

Reviewer 3 Report

1. The first paragraph of the introduction lacks sufficient elaboration and fails to provide a clear background for the research problem. References should be added to support the statements.

2. This study builds upon previous research, but the level of advancement in the previous work and the specific contributions of this study need to be addressed. A comparison with state-of-the-art methods should be provided, rather than solely comparing it to the author's previous work.

3. Line 147: The novelty of this study is not adequately explained. It is crucial to specify the specific shortcomings of previous methods in detail.

4. Line 170: While mentioning that this study does not require special hardware and achieves fast speed, the introduction should emphasize the methods employed to achieve these results, as it is a scientific research article.

5. The connection between the introduction and the literature review in the second part is weak. It requires revision to enhance readability and cohesiveness.

6. It is important to compare and analyze the performance of the proposed method against the current state-of-the-art method in the experiments.

7. Future work section can include emerging deep learning methods such as implicit functions and attention mechanisms as lightweight enhancement representation algorithms.
[1] Chu, H., Long, L., Guo, J., Yuan, H. and Deng, L., 2023. Implicit function‐based continuous representation for meticulous segmentation of cracks from high‐resolution images. Computer‐Aided Civil and Infrastructure Engineering.
[2] Chu, H., Wang, W. and Deng, L., 2022. Tiny‐Crack‐Net: A multiscale feature fusion network with attention mechanisms for segmentation of tiny cracks. Computer‐Aided Civil and Infrastructure Engineering, 37(14), pp.1914-1931.

8. The language used in the article needs improvement to adhere to academic standards.

1. Line 8, "Most of the...". "of the" should be removed to ensure brevity in scientific writing. 
2. Line 11, ".... Estimation (FME) part, is an extra.....". Wrong sentence, "," needs to be removed to ensure that the composition of the sentence is correct.
3. Line 14, "...
that makes also..." should be changed to "...that also makes...".
4. 
Line 14, "...are shown to provide...". Wrong sentence.

The aforementioned listed issues are just a part of the overall similar issues present in the entire text. The authors are requested to revise the entire text accordingly.

Author Response

Response to Reviewer 3 Comments

Thank you for the valuable feedback and your comments.

Point 1: 1. The first paragraph of the introduction lacks sufficient elaboration and fails to provide a clear background for the research problem. References should be added to support the statements.

Response 1: We rewrote the Introduction section.

Point 2: This study builds upon previous research, but the level of advancement in the previous work and the specific contributions of this study need to be addressed. A comparison with state-of-the-art methods should be provided, rather than solely comparing it to the author's previous work.

Response 2: Despite that for completeness reasons, we provided some state-of-the-art hardware solutions (Refs. 16-22), they belong to a different category from what we focused on in most of our work because a) they use custom hardware-friendly FME algorithms and b) they aim mostly to companies and professionals with real-time encoding needs and high costs. Although, in the “Related Work” section we tried to make performance comparisons, stating also our conclusions. 

Point 3: Line 147: The novelty of this study is not adequately explained. It is crucial to specify the specific shortcomings of previous methods in detail.

Response 3: We rewrote it, to make it more clear for you (lines 275-292)

Point 4: Line 170: While mentioning that this study does not require special hardware and achieves fast speed, the introduction should emphasize the methods employed to achieve these results, as it is a scientific research article.

Response 4: We rewrote our introduction after your suggestion.

Point 5: The connection between the introduction and the literature review in the second part is weak. It requires revision to enhance readability and cohesiveness.

Response 5: We revised also the Related Work.

Point 6: It is important to compare and analyze the performance of the proposed method against the current state-of-the-art method in the experiments.

Response 6: In the “Related Works” section, we tried to compare different categories implementations  (hardware vs software) with different algorithms (FME optimized algorithms vs FME skipped algorithms) but this is not always easy since we are not using the same hardware setup (FPGAs, CPU cores and clock speed, Memory, GPU generation etc), and most importantly the same video sequences to compare with. We used common Class B and A video sequences (table 1) which conform with the guidelines of the “Common Test Conditions and Software Reference Configurations” from the JCT-VC video experts group. Also, all of our experiments are conducted in an innovative Multi CPU/GPU environment which was not the case for most of the software solutions. For that reason,  experiments include comparison tests with the default non-modified HEVC encoder (1 thread, sequential) in the same hardware setup as described in the “4.2 Setup” section to allow the researchers to analyze and compare the performance with their own setups.

Point 7: Future work section can include emerging deep learning methods such as implicit functions and attention mechanisms as lightweight enhancement representation algorithms.

[1] Chu, H., Long, L., Guo, J., Yuan, H. and Deng, L., 2023. Implicit function‐based continuous representation for meticulous segmentation of cracks from high‐resolution images. Computer‐Aided Civil and Infrastructure Engineering.

[2] Chu, H., Wang, W. and Deng, L., 2022. Tiny‐Crack‐Net: A multiscale feature fusion network with attention mechanisms for segmentation of tiny cracks. Computer‐Aided Civil and Infrastructure Engineering, 37(14), pp.1914-1931

.

Response 7: Thank you for your recommendations. We added a reference at lines 334-340

Point 8: The language used in the article needs improvement to adhere to academic standards.

//

Response 8: We revised our text. MDPI editors will also review the final text for such issues. Thank you.

Comments on the Quality of English Language

  1. Line 8, "Most of the...". "of the" should be removed to ensure brevity in scientific writing.
  2. Line 11, ".... Estimation (FME) part, is an extra.....". Wrong sentence, "," needs to be removed to ensure that the composition of the sentence is correct.
  3. Line 14, "...that makes also..." should be changed to "...that also makes...".
  4. Line 14, "...are shown to provide...". Wrong sentence.

The aforementioned listed issues are just a part of the overall similar issues present in the entire text. The authors are requested to revise the entire text accordingly.

Reviewer 4 Report

The proposed method in this manuscript aims to improve the encoding speed by skipping some fraction motion estimations (FMEs) based on a motion vector or a distance displacement calculated through integer motion estimations (IMEs). However, the increases in both bit-rate and image distortion compared to the standard video encoding appear to be non-negligible. This point is concerned because it seems to be different from the purpose of image encoding, which compresses images while preserving image quality. The authors need to clarify the purpose of applying the proposed method more clearly.

  1. - Please clearly explain why it is necessary to increase the video encoding speed while accepting the increase in video distortion and bit rate in the Introduction section, i.e., describe fields to which the proposed method is applied in detail.

    - In Abstract, a criteria to skip FME should be described in more detail.

    - The proposed method should be compared to WTP+IME in terms of bit-rate, image distortion, and video encoding speed.

    - In Figure 5-6, 9, the units of the y-axis are ambiguous. It seems to be the speed-up rate.

Author Response

Response to Reviewer 4 Comments

Thank you for the valuable feedback and your comments.

The proposed method in this manuscript aims to improve the encoding speed by skipping some fraction motion estimations (FMEs) based on a motion vector or a distance displacement calculated through integer motion estimations (IMEs). However, the increases in both bit-rate and image distortion compared to the standard video encoding appear to be non-negligible. This point is concerned because it seems to be different from the purpose of image encoding, which compresses images while preserving image quality. The authors need to clarify the purpose of applying the proposed method more clearly.

Point 1: - Please clearly explain why it is necessary to increase the video encoding speed while accepting the increase in video distortion and bit rate in the Introduction section, i.e., describe fields to which the proposed method is applied in detail.

Response 1:  As we stated multiple times in our revised version (Introduction section lines 23-25, Coding Efficiency Results section lines 458-466, and Conclusions section lines 519-523) the negative effects of the WTP Tiling scheme can be deteriorated by using a lower Tiling scheme pattern. So, it's up to the user the select between speed (higher Tiling scheme) or quality (lower Tiling scheme pattern) according to his needs. The FER algorithm's negative effects are negligible (Fig. 11 and Fig 12). The Bit-Rate and PSNR increment is more related to the way Context-adaptive binary arithmetic coding (CABAC) works internally and the WTP pattern selection scheme. Using FER with a third-party encoder no considerable drop quality is expected. WTP+GPU encoder pros and cons are analyzed and explained in our previous works in more detail (Refs 9-10). Here we are using the WTP-GPU encoder as a tool to apply and demonstrate the gain of FER which is a generic skipping algorithm and as already mentioned, it can be applied with any other encoder.

Point 2: In Abstract, a criteria to skip FME should be described in more detail.

Response 2: We added this according to your suggestion.

Point 3: The proposed method should be compared to WTP+IME in terms of bit rate, image distortion, and video encoding speed

Response 3: Video encoding speed differences are already depicted in Fig 10 (WTP+IME= is the same terminology as WTP+GPU). After your suggestions, we also added Fig. 11 (PSNR) and Fig. 12 (BitRate) differences.

Point 4: In Figure 5-6, 9, the units of the y-axis are ambiguous. It seems to be the speed-up rate.

Response 4: Yes, that’s correct, they are speed-up rates. Y-axis is denoted as “SPEEDUP” as well as is clearly stated in the captions under the mentioned plots.  

Reviewer 5 Report

Authors propose a fractional execution resolver using a hybrid multi-CPU/GPU 2 encoding scheme. To improve the quality of their manuscript, my comments and suggestions are listed as follows.

(1) The word “a” in the title can be removed.

(2) For the Section Abstract, the conclusion content is missing. Authors should use one or more sentences to show their conclusions in this manuscript.

(3) There are lots of keywords. It is usually less than 8 phrases. Meanwhile, for the abbreviations, the full spellings are also expected.

(4) The paper architecture is expected to show the content of each section in brief as the last paragraph of Section Introduction.

(5) Some paragraphs are very long. Authors are suggested to make the paragraph less than 10-line text.

(6) The pseudocode writing is not very standard. The input and output values are expected.

(7) The content of Section conclusion. can be concise. It is too long at present.

(8) Most of references are very old. Authors should make the literature survey again and update them.

There are some typos, so minor editing of English language is required.

Author Response

Response to Reviewer 5 Comments

Thank you for the valuable feedback and your comments.

Point 1: The word “a” in the title can be removed.

Response 1: It was removed.

Point 2: For the Section Abstract, the conclusion content is missing. Authors should use one or more sentences to show their conclusions in this manuscript.

Response 2: The conclusion content is briefly stated in lines 19-28 of the Abstract Section. Specifically, it states: «Our evaluations are shown to provide …..». We added some extra conclusions in the Abstract section, according to your recommendations.

Point 3: There are lots of keywords. It is usually less than 8 phrases. Meanwhile, for the abbreviations, full spellings are also expected.

Response 3: Keywords are reduced to five (5) and full spelling abbreviations are construed. Although, the already construed abbreviations in previous sections are not explained again.

Point 4: The paper architecture is expected to show the content of each section in brief as the last paragraph of the Section Introduction.

Response 4: We tried to add some introduction info at the beginning of every section

Point 5: Some paragraphs are very long. Authors are suggested to make the paragraph less than 10-line text.

Response 5: We could not detect paragraphs with more than 10 text lines. Would you be kind enough to pinpoint that paragraphs? 

Point 6: The pseudocode writing is not very standard. The input and output values are expected.

Response 6: Except for the “isStillBlock” variable which is a newly defined variable, the rest of them are controlled by the encoder and the TZS algorithm itself. For them, we kept the original TZS name definition and functionality, that they are both well-defined from the references [3,7] and the HEVC standard itself.  Βriefly, BestX, and BestY are the initial best motion vectors offsets, loaded from the prediction part of the encoder, iDist is the distance from the center of the search point with an initial value of 1, iStartX, iStartY is the new search point in every cycle initialized from BestX and BestY values at the beginning, uiBestSAD is the current Best SAD for every search cycle calculated internally from the current iStartX,iStartY values, uiBestRound is the loop search cycle counter updated also internally, and uiBestDistance pinpoints the best candidate distance so far from the center of the search area. The “ui” prefix before some variables, stands for unsigned integer value.

Point 7: The content of Section conclusion. can be concise. It is too long at present.

Response 7: Other reviewers asked for the conclusion to be more of an afterthought. They recommended we highlight important findings and include post hoc reflections on this work and future perspectives.

Point 8: Most of references are very old. Authors should make the literature survey again and update them.

Response 8: Except for the guides or the language standards i.e. OpenCL, Cuda Programming, etc, that are evolving slowly with minor updates and additions,  we tried to keep an up-to-date reference list. We tried to add the most recent publications, but honestly, it seems that there are not many updates to this specific topic except for some hardware proposals. We updated our Reference list, but we focused more on software solutions.

Round 2

Reviewer 1 Report

Resolved all the issues raised

Just few minor edit 

Author Response

Your contributions are greatly valued.  We appreciate the information and advice you have shared—many thanks for your assistance.

Reviewer 3 Report

All of the reviewer's inquiries have been addressed satisfactorily. Suggest  to accept in present form.

Author Response

(The authors gave the same response as above.)

Reviewer 4 Report

The author's response solved the concerns about the proposed method. There are minor modifications as follows.

1. The Introduction Section is too long. The detailed descriptions about the problems of the existing FME and the proposed method need to be moved to new sections or be merged in existing sections.
2. The image qualities of the equations and the variables are poor. It is necessary to increase the image quality or to replace the equations and the variables using the formula editing features supported by the editor (MS Word or LaTeX).

Author Response

Point 1. The Introduction Section is too long. The detailed descriptions about the problems of the existing FME and the proposed method need to be moved to new sections or be merged in existing sections.

Response 1: We moved the detailed descriptions of the pFTDP function inside the "Implementation" section (lines 370-384)

Point 2. The image qualities of the equations and the variables are poor. It is necessary to increase the image quality or to replace the equations and the variables using the formula editing features supported by the editor (MS Word or LaTeX).

Response 2: Due to an uncoverable error in the original ms word file we were forced to save the draft in an older MS Office version causing degradation of the math symbols. We restored our original version and now are all set in their original vector form.

Reviewer 5 Report

Authors have carefully revised their manuscript according to my comments and suggestions. The quality of this manuscript is improved obviously. However, authors should pay attention to the following issues before the formal publication.

(1) The content of Section conclusion can be concise. It is too long at present.

(2) The pseudocode writing is not very standard. Authors should point out the input and output values.

(3) There are some paragraphs are very long, such as the paragraph in the line 222 to 258, and the paragraph in the line 260 to 304.

Author Response

Your contributions are greatly valued. We appreciate the information and advice you have shared—many thanks for your assistance.

Point 1: The content of Section conclusion can be concise. It is too long at present.

Response 1: We tried to keep a balance among the reviewer's suggestions.

Point 2: The pseudocode writing is not very standard. Authors should point out the input and output values.

Response 2: We redesigned the pseudocode, as you suggested. Thank you.

Point 3: There are some paragraphs are very long, such as the paragraph in the line 222 to 258, and the paragraph in the line 260 to 304.

Response 3: Thank you, we added some breaks after your suggestion.